# Exosomal Small RNA Sequencing Profiles in Plasma from Subjects with Latent *Mycobacterium tuberculosis* Infection

**DOI:** 10.3390/microorganisms12071417

**Published:** 2024-07-12

**Authors:** Xiaogang Cui, Hangting Meng, Miao Li, Xia Chen, Dan Yuan, Changxin Wu

**Affiliations:** Key Lab of Medical Molecular Cell Biology of Shanxi Province, Institutes of Biomedical Sciences, Shanxi University, Taiyuan 030006, China; 202223119015@email.sxu.edu.cn (H.M.); 202323119018@email.sxu.edu.cn (M.L.); 202123105002@email.sxu.edu.cn (X.C.); 202323105014@email.sxu.edu.cn (D.Y.)

**Keywords:** latent tuberculosis infection (LTBI), exosomes, miR-7850-5p, SLC11A1

## Abstract

Despite huge efforts, tuberculosis (TB) is still a major public health threat worldwide, with approximately 23% of the human population harboring a latent TB infection (LTBI). LTBI can reactivate and progress to active and transmissible TB disease, contributing to its spread within the population. The challenges in diagnosing and treating LTBI patients have been major factors contributing to this phenomenon. Exosomes offer a novel avenue for investigating the process of TB infection. In this study, we conducted small RNA sequencing to investigate the small RNA profiles of plasma exosomes derived from individuals with LTBI and healthy controls. Our findings revealed distinct miRNA profiles in the exosomes between the two groups. We identified 12 differentially expressed miRNAs through this analysis, which were further validated via qRT-PCR using the same exosomes. Notably, six miRNAs (hsa-miR-7850-5p, hsa-miR-1306-5p, hsa-miR-363-5p, hsa-miR-374a-5p, hsa-miR-4654, has-miR-6529-5p, and hsa-miR-140-5p) exhibited specifically elevated expression in individuals with LTBI. Gene ontology and KEGG pathway analyses revealed that the targets of these miRNAs were enriched in functions associated with ferroptosis and fatty acid metabolism, underscoring the critical role of these miRNAs in regulating the intracellular survival of *Mycobacterium tuberculosis* (*Mtb*). Furthermore, our results indicated that the overexpression of miR-7850-5p downregulated the expression of the SLC11A1 protein in both *Mtb*-infected and *Mtb*-uninfected THP1 cells. Additionally, we observed that miR-7850-5p promoted the intracellular survival of *Mtb* by suppressing the expression of the SLC11A1 protein. Overall, our findings provide valuable insights into the role of miRNAs and repetitive region-derived small RNAs in exosomes during the infectious process of *Mtb* and contribute to the identification of potential molecular targets for the detection and diagnosis of latent tuberculosis.

## 1. Introduction

*Mycobacterium tuberculosis* (*Mtb*) is a significant human pathogen capable of causing pulmonary disease with the potential to disseminate to various organs, presenting as an acute, chronic, or latent infection [1]. Latent tuberculosis infection (LTBI) is characterized by an enduring immune response to *Mtb* antigens without active tuberculosis (TB) manifestations, as defined by the World Health Organization (WHO) [2]. Accurate diagnosis of latent TB is currently not achievable through a single test. Diagnosis of LTBI relies on detecting cellular immune responses to *Mtb* while reasonably excluding active TB [3,4]. In low-incidence settings, public health strategies focus on identifying and treating LTBI to prevent potential reactivation leading to active and transmissible TB [5]. Therefore, early diagnosis of LTBI patients is crucial for reducing morbidity and preventing onward transmission to vulnerable individuals.

In recent years, exosomes, small extracellular vesicles originating from diverse cell types, have demonstrated significant potential as diagnostic biomarkers and therapeutic agents based on their cargo composition [6,7]. Exosomes contain characteristic proteins such as tetraspanin proteins CD9, CD63, and CD81, which are commonly located on exosome membranes and are involved in membrane fusion, signaling, and protein trafficking [8]. Exosome biogenesis also involves ALIX, flotillin, and TSG101. MVB formation and ILV engulfment are facilitated by ESCRT and its associated proteins Hrs, flotillin, TSG101, and ALIX, which are frequently found in exosome cargo [8]. Notably, exosomal microRNAs (miRNAs) have emerged as pivotal regulators in the immune response to *Mtb* infection [9]. Consequently, exosomes offer a promising avenue for TB research, with studies revealing that exosomal miRNAs modulate host defenses against *Mtb* by influencing key signaling pathways during infection [7,10]. Moreover, increasing evidence suggests that exosomes facilitate the transfer of mRNAs, enabling the exchange of phenotypic traits between cells and identifying potential TB biomarkers [10,11]. Nevertheless, comprehensive exosomal RNA sequencing analysis in human clinical samples from LTBI individuals is scarce.

In this study, we conducted a comprehensive analysis of exosomal RNA profiles from the plasma samples of individuals with LTBI and healthy controls (HCs). Our findings demonstrated distinct gene expression signatures within exosomes, suggesting that the Exo package reflects what is more highly present in the cell. Additionally, we utilized an in vitro model to mimic *Mtb* infection in THP1 macrophages and successfully validated the expression of miR-7850-5p, a candidate miRNA identified through screening. These results offer valuable insights into exosome dynamics during *Mtb* infection and shed light on the potential utility of exosomal RNAs as biomarkers for LTBI diagnosis.

## 2. Materials and Methods

### 2.1. Patient Information and Sample Preparation

A total of 50 individuals were screened and categorized into two groups: a healthy control (HC) group consisting of 34 individuals and an LTBI group consisting of 16 individuals. Fresh whole-blood samples of 5 mL were collected in EDTA-coated anticoagulant tubes from the HCs and patients with LTBI. The samples were promptly centrifuged at 4 °C and 3000× *g* for 15 min to isolate the plasma. The plasma samples were then combined into pooled samples, with 8 individuals in each HC pool and 4 individuals in each LTBI pool (1 mL of plasma per pooled sample). Four biological replicate pools were prepared for each group. The participant demographic information is detailed in Table 1.

An informed consent form, in compliance with the Declaration of Helsinki, was signed by all participants, and the study protocol received approval from the Ethics Committee of Shan Xi University. The cohort comprised HIV-negative adults aged 18 years or older with no history of tuberculosis. LTBI was diagnosed in individuals by positive tuberculin skin test (TST) and interferon-gamma release assay (IGRA) results. The HC subjects were characterized by normal chest computed tomography (CT) scans and negative TST and IGRA outcomes.

### 2.2. Exosome Isolation and RNA Extraction

Exosomes were isolated from the plasma using differential ultracentrifugation as previously described [12]. The isolated exosomes were measured via nanoparticle tracking analysis (NTA) using ZetaVIEW^®^ equipment (Particle Metrix, Munich, Germany). The morphological characteristics were observed using JEM-1400 TEM (JEOL, Tokyo, Japan). The procedure for preparing the samples for TEM was as follows: exosome pellets were resuspended and deposited onto 200-mesh formvar carbon-coated nickel grids, then incubated with 50 mM glycine. Following blocking with 5% bovine serum albumin (BSA), the specimens were treated with rabbit anti-human antibodies (anti-CD9 (SBI, Los Angeles, CA, USA), anti-CD81 (SBI, USA), anti-TSG101 (Abcam, Waltham, MA, USA), anti-calnexin (Abcam, USA), and anti-β-actin (Abclonal, Wuhan, China)). Subsequently, the samples were exposed to a goat anti-rabbit secondary antibody conjugated with protein A-gold particles (10 nm) (Bioss, Beijing, China), and subjected to negative staining with 3% phosphotungstic acid for 10 min [13].

The isolated exosomes were immediately used for total RNA extraction using RNAiso-Plus (TaKaRa, Dalian, China) according to the manufacturer’s instructions. RNA concentration was measured using a Qubit^®^ RNA Assay Kit in a Qubit^®^ 2.0 Fluorometer (Life Technologies, Camarillo, CA, USA). RNA integrity was assessed using the RNA Nano 6000 Assay Kit of the Agilent Bioanalyzer 2100 system (Agilent Technologies, Santa Clara, CA, USA) [14].

### 2.3. Small RNA-Seq and Data Analysis

Small RNA libraries were prepared, and RNA sequencing (RNA-Seq) was conducted by BGI Genomics (Wuhan, China). Small RNAs ranging from 18 to 30 nucleotides in size were isolated using PAGE and ligated separately to 3′ and 5′ adapters. Next, the cDNAs derived from these purified small RNAs were synthesized through reverse transcription and amplified using PCR. The resulting PCR products underwent purification via PAGE and were denatured into single-stranded DNA.

Single-stranded cyclized DNA molecules were generated through a circularization reaction and amplified into DNA nanoballs (DNBs) using rolling cycle amplification. High-quality DNBs were loaded onto patterned nanoarrays using a high-intensity DNA nanochip technique and sequenced via combinatorial probe-anchor synthesis. To ensure data quality, we filtered the raw sequencing data to remove low-quality tags, tags lacking a 3′ primer or insertion, tags contaminated with 5′ primers, poly-A, and tags shorter than 15 nucleotides to obtain clean tags. These clean tags were aligned to the reference genome and various small RNA databases (e.g., siRNA, miRBase, piRNA, and snoRNA) using Bowtie2-2.5.2 (Langmead et al., 2009) [15]. Differentially expressed miRNAs (DEmiRNAs) were identified using DESeq2 (Love et al., 2014) [16] within the Dr. Tom Multi-omics Data Mining System (https://biosys.bgi.com, accessed on 17 June 2024) developed by BGI. A significance threshold of *q* < 0.05 was applied to assess differential expression. Furthermore, target genes of validated DEmiRNAs were predicted using TargetScan and miRanda, and their molecular functions and pathways were analyzed using GO (http://www.geneontology.org/, accessed on 17 June 2024) and KEGG analyses (https://www.genome.jp/kegg/, accessed on 17 June 2024).

### 2.4. Real-Time Quantitative PCR (RT-qPCR)

mRNA and DEmiRNA expression profiling was conducted using the LightCycler 480 II Real-Time PCR System (LightCycler, Indianapolis, IN, USA) with a MonAmp^TM^ Universal ChemoHS Specificity Plus qPCR Mix (Monad, Wuhan, China) and a miRcute Plus miRNA qPCR Kit (SYBR Green) (Tiangen, Beijing, China). The qPCR protocol included an initial denaturation step at 95 °C for 15 min, followed by 45 cycles at 94 °C for 20 s and 60 °C for 34 s. All experiments were performed in triplicate.

The mRNA and miRNA expression levels were normalized to β-actin and U6, respectively. Relative quantification of the target genes and miRNAs was determined using the 2^−ΔΔCt^ method.

### 2.5. Western Blot Analysis

Western blot analysis was employed to assess the protein-level expression of key molecules associated with SLC11A1 in Phorbol-12-myristate-13-acetate (PMA)-differentiated THP-1 macrophages following the overexpression of miR-7850-5p via transfection with a miR-7850-5p mimic or the downregulation of miR-7850-5p using a miR-7850-5p inhibitor.

The transfected cells underwent lysis using a lysis buffer (Sigma, College Park, MD, USA). Following this, the lysates were subjected to separation via 10% SDS-PAGE and then transferred onto a polyvinylidene difluoride (PVDF) membrane (Millipore, Burlington, MA, USA). Subsequently, the membranes were blocked using 5% BSA in TBST (TBS with Tween 20) and incubated overnight at 4 °C with the primary antibodies anti-SLC11A1 (Abcam, ab211448, 1:1000 dilution) and anti-β-actin (Abclonal, Wuhan, China, AC028, 1:1000 dilution). This was followed by incubation with specific horseradish peroxidase-conjugated secondary antibodies (either goat anti-rabbit IgG or goat anti-mouse IgG (Bioss, Beijing, China, 1:5000 dilution)) for 1 h at 37 °C. Finally, protein bands were visualized through chemiluminescence utilizing SuperKing™ Hypersensitive luminescent ELC solution (Abbkine, Beijing, China). Quantitative analysis of the bands was carried out using ImageJ x64 1.8.0 software.

### 2.6. Plasma-Derived Exosome Uptake by THP1 Cells

Exosomes isolated from the plasma samples of both healthy individuals and LTBI patients were co-incubated with THP1 cells at 37 °C for 12 h. Then, the expression of the protein SLC11A1 was tested using Western blotting.

### 2.7. In Vitro Cell Culture Model under Mtb-Specific Antigen Stimuli

*Mtb* H37Ra was provided by Prof. Qian Gao (Fudan University, Shanghai, China). The THP1 cell line was cultured according to ATCC guidelines, utilizing RPMI 1640 (Gibco, New York State, NY, USA) supplemented with 10% FBS (Hyclone) and 1x penicillin (stock concentration = 100 U/mL) and streptomycin (stock concentration = 100 μg/mL). The cells were maintained in complete media at 37 °C in a humidified atmosphere with 5% CO_2_.

THP1 cells were differentiated into macrophage-like cells by treating them with 5 ng/mL of PMA for 48 h. A total of 30 million THP1 cells per T-175 flask were differentiated in multiple flasks to cover all time points and strains. THP1 cells were infected with H37Ra at a multiplicity of infection (MOI) of 10:1 (bacteria-to-cell ratio) for 10 h (P^2+^ biosafety level). Subsequently, the infected cells were washed three times with PBS in preparation for subsequent experiments. Transfected THP1 cells were infected with *Mtb* at an MOI of 10 for 10 h and then lysed using 0.1% Triton-X 100 (Solarbio, Beijing, China). Serial dilutions of 10-fold and 100-fold were prepared and utilized for quantitative culture. Subsequently, 60 μL of each dilution was plated on Middlebrook 7H10 agar plates (BD, Franklin Lakes, NJ, USA) supplemented with 10% OADC enrichment (BD, USA). Following a 2-week incubation, *Mtb* colonies were observed, and colony-forming units (CFUs) were calculated.

### 2.8. Transfection of miR-7850-5p Mimic/Inhibitor in THP1 Cells

miR-7850-5p mimic and inhibitor at a final concentration of 50 nM, along with a miR-nonsense sequence control (NC), were synthesized by GenePharma (Shanghai, China) and transfected into THP-1 cells using Lipofectamine 3000 (Invitrogen, Carlsbad, CA, USA) for 48 h. miR-7850-5p mimics are double-stranded small RNA molecules designed based on the mature miRNA sequence and used to mimic the endogenous mature miRNA sequence, and miR-7850-5p inhibitors are a tool used to suppress the function of endogenous miRNA. miR-7850-5p and SLC11A1 mRNA levels were quantified using qRT-PCR, while SLC11A1 protein expression was assessed via Western blotting.

### 2.9. Statistical Analysis

Statistical analyses were conducted using GraphPad Prism 5 (GraphPad Software Inc., San Diego, CA, USA). Normal variables were assessed using the two-tailed Student’s *t*-test and presented as mean ± standard deviation, while non-normally distributed variables were analyzed using nonparametric tests and reported as median with interquartile range. Spearman’s correlation analysis was employed to examine relationships between continuous variables.

## 3. Results

### 3.1. Patient Characteristics

The recorded patient demographics were age, gender, smoking/drinking habits, educational background, and marital status. Table 1 shows that among the participants in the latent pulmonary TB group with positive IGRA results, there were 4 males and 12 females. Among the healthy individuals in the control group, there were 2 males and 32 females. The IGRA findings indicated a significant difference between the two groups in terms of sex (*p* ≤ 0.01) (Table 1).

### 3.2. Isolation and Characterization of Plasma-Derived EVs

We performed a comprehensive analysis of the isolation and characterization of exosomes, as well as small RNA sequencing of human plasma-derived EVs from individuals with LTBI and HCs. The initial exosome isolation and characterization procedures entailed a differential ultracentrifugation. Plasma-derived exosomes ranging in size from approximately 50 to 150 nm were examined using TEM and NTA. The TEM analysis validated the cup-shaped morphology of the plasma-derived exosomes (Figure 1B,D). Western blotting analysis was employed to evaluate the presence of exosome markers in plasma-derived EVs from individuals with LTBI and HCs, revealing an enrichment of exosomal markers CD9, CD81, and TSG101. Importantly, exosomes isolated from human plasma samples were found to be free of calnexin, indicating the absence of endoplasmic reticulum protein contamination (Figure 1C). In conclusion, pure exosomes were obtained via differential ultracentrifugation, laying the foundation for subsequent proteomics (Figure 1A).

### 3.3. Expression Panels of the Plasma Exosomes in the HC and LTBI Groups

The data quality control and comparative analysis results are presented in Appendix A. The raw tag counts were 37, 748, and 736 in each sample. Clean data (clean tag counts) were obtained by excluding reads containing “N” and reads with a length <18 or >30. The proportion of clean tag counts was above 80.89%. More than 98.4% of the clean tag counts had Q20 (the proportion of bases with a Phred base quality score greater than 20, i.e., the proportion of read bases whose error rate is less than 1%). Based on the CPM (counts per million using the following formula: normalized expression = mapped read count/total reads × 1,000,000)-mapped fragment of each sample, the correlation coefficient (R^2^) among the four pools or samples was determined assuming 16 individuals in the LTBI group as explained in Section 2.1. This indicated a high level of similarity among the four biological replicates within each group (Figure 2A).

To detect changes in miRNA expression in the plasma exosomes, a small RNA-Seq analysis was conducted using plasma exosomal miRNA samples isolated from four mixed plasma samples from the patients with LTBI and four mixed plasma samples from the healthy controls. As demonstrated in the volcano plot (Figure 2B), 12 of the identified miRNAs exhibited differential expression, with 6 downregulated (hsa-let-7e-3p, hsa-miR-1283, hsa-miR-142-5p, hsa-miR-522-3p, hsa-miR-582-5p, and hsa-let-7a-5p) and 6 upregulated (hsa-miR-7850-5p, hsa-miR-1306-5p, hsa-miR-363-5p, hsa-miR-374a-5p, hsa-miR-4654, and hsa-miR-6529-5p) in LTBI patients (Figure 2C). The mature sequences and precursors of 12 DEmiRNAs are presented in Appendix A.

### 3.4. Transfer of Plasma-Derived Exosomes to THP1 Cells

To elucidate the mechanism of miR-7850-5p intercellular delivery, co-culture experiments were performed to determine whether exosomes and their contents could be internalized by THP1 cells after 12 h. The results indicated the entry of numerous exosomes into the THP1 cells accumulating around the nucleus after 12 h of co-cultivation. Further examination of the effects of plasma-derived exo-miR-7850-5p on the expression of SLC11A1 in THP1 cells involved isolating protein from co-cultured THP1 cells. A Western blot analysis demonstrated that the expression of SLC11A1 was inhibited upon incubation with exo-miR-7850-5p in THP1 cells, whereas the inhibition of miR-7850-5p promoted the expression of SLC11A1 (Figure 2D).

### 3.5. Bioinformatics Analyses of Exosomes in LTBI Patients

Based on the TargetScan and miRanda databases, a network diagram illustrating the interaction between differentially expressed miRNAs and target genes in exosomes was generated using Cytoscape 3.10.0 software (Appendix A). This diagram effectively depicts the interaction relationships of diverse miRNAs (Figure 3A).

The analysis utilizing Gene Ontology (GO) categorized the outcomes based on biological processes (BPs), cellular components (CCs), and molecular functions (MFs) (Appendix A). For BPs, the target genes primarily participated in transcriptional regulation, protein phosphorylation, and other activities. Regarding MFs, the exosomal mRNA displayed enrichment in molecules with molecular binding properties. Concerning CCs, the exosomal mRNA exhibited enrichment in lysosomes, endosomes, and plasma membranes (Figure 3B). The KEGG pathway analysis indicated the enrichment of target genes in pathways such as ferroptosis, MAPK signaling, and fatty acid metabolism (Appendix A) (Figure 3C).

These analyses suggest that the identified target genes of the exosomal miRNAs will serve as a valuable resource for exosome research and underscore the necessity for further experimental assessment of the functions of these identified exosomal miRNAs in the physiology and pathophysiology of individuals with LTBI.

### 3.6. Validation of DEmiRNAs Using Quantitative Real-Time PCR (qRT-PCR)

A comparison of the expression of these 12 miRNAs was performed between the qRT-PCR data normalized to U6 and those derived from small RNA-Seq. Strong concordance in expression patterns between small RNA-Seq and qRT-PCR was observed, which confirmed the reliability of the statistical criteria used in these analyses (Figure 4).

### 3.7. Expression Levels of SLC11A1 Regulated by miR-7850-5p in THP1 Cells

To evaluate the effects of miR-7850-5p on SLC11A1 expression, we transfected miR-7850-5p mimics and their inhibitors into *Mtb*-uninfected and *Mtb*-infected THP1 cells. The results showed that miR-7850-5p mimics reduced SLC11A1 expression, while their inhibitors displayed the opposite effects, both in the *Mtb*-uninfected (Figure 5A,B) and *Mtb*-infected THP1 cells (Figure 5C,D). Furthermore, the colony-forming unit counting assay demonstrated that miR-7850-5p increased intracellular *Mtb* survival, whereas the miR-7850-5p inhibitor decreased intracellular *Mtb* survival (Figure 5E). Collectively, our results indicate that miR-7850-5p affects the intracellular survival of *Mtb* in THP1 by targeting SLC11A1.

## 4. Discussion

Extracellular vesicles (EVs) and exosomes are known to carry proteins, mRNAs, and non-coding RNAs (ncRNAs) and play a key role in mediating various cellular and biological processes through intercellular communication between adjacent and distant cells [17,18,19,20]. Recent analyses of small RNA sequences in different body fluids, including blood, leukocytes, plasma, serum, saliva, cell-free saliva, urine, and cell-free urine have revealed the presence of distinct miRNAs, tRNAs, and piRNAs [12]. The proteins in EVs can participate in fundamental cellular processes, such as cell adhesion, structural dynamics, membrane fusion, metabolism, and signal transduction [21]. EV-derived miRNAs can be delivered to target cells and regulate mRNA function. EVs can be secreted by the majority of mammalian cells and play a role in intercellular communication and dissemination. Studies have shown that EVs can induce the secretion of cytokines [22] and tumor necrosis factor [23] by macrophages, enhance the activity of natural killer cells, and facilitate antigen presentation [24]. EVs also play a role in neural communication, with miRNAs from hypothalamic neural stem cell-derived EVs capable of attenuating the aging process [25]. Moreover, EVs contribute to various reproductive and developmental processes, including gamete maturation, fertilization, and embryo implantation [25]. In this study, we conducted a comparative analysis of small RNAs based on RNA sequencing analysis of human plasma-derived EVs, focusing on ncRNAs, specifically miRNAs that were differentially expressed in LTBI patients compared to HCs. Twelve differently expressed miRNAs were identified; among these miRNAs, six were highly expressed in LTBI plasma-derived exosomes and six were lowly expressed relative to the healthy control group.

To further explore the mechanism of RNA packaging into exosomes under an infectious status, we performed functional and pathway analyses. Approximately 57 functional items were identified in the LTBI group compared to the HC individuals, including “axon guidance”, “AMPK signaling pathway”, “ferroptosis”, and “mTOR signaling pathway”. Interestingly, the KEGG pathway analyses suggested that SLC11A1 is involved in the AMPK signaling pathway, Ras signaling pathway, ferroptosis, MAPK signaling pathway, and fatty acid metabolism. Ferroptosis is a lipid peroxidation-driven and iron-dependent programmed cell death involved in multiple physical processes and various diseases [26]. Recent research has indicated that the inhibition of iron death in macrophages reduces the bacterial burden and tissue necrosis caused by *Mtb* [27]. Furthermore, clinical studies have demonstrated that TB patients receiving treatment with anti-iron death drugs, such as vitamin E or selenoenzyme, show improved treatment outcomes [28,29].

The SLC11A1 protein in macrophages and neutrophils is crucial for fighting pathogens like *Mtb* [30]. It helps by moving certain metals in cells, starving bacteria, and making the environment hostile toward them [31,32,33]. In mice, a functional Slc11a1 gene means survival from infections, while a faulty gene leads to deadly infections. Low SLC11A1 levels increase the risk of TB [34,35]. Furthermore, SLC11A1 facilitates intracellular iron sequestration by mediating its translocation from the phagolysosome to the cytoplasmic compartment, thereby reducing the bioavailability of the metal for pathogens such as *Mtb* and restricting their proliferation [36,37,38,39]. Here, SLC11A1 was predicted as a target gene for miR-7850-5p. In addition, the results of the current study showed that LTBI patients’ plasma exosomes carried miR-7850-5p to THP1, inhibiting the key iron death gene SLC11A1′s expression and promoting the intracellular survival of *Mtb* in THP1. Based on this, we hypothesize that miR-7850-5p promotes macrophage iron death by targeting SLC11A1, thereby exacerbating the bacterial burden and tissue necrosis caused by *Mtb*. However, the specific molecular mechanism upregulating miR-7850-5p during the process of *Mtb* infection needs to be studied further.

In the initial phase of tuberculosis, macrophages elicit a signature miRNA response during *Mtb* infection, detectable within plasma-derived exosomal fractions [40,41]. Differential miRNA expression modulates intracellular signaling cascades implicated in the infection process [41]. miRNA profiling elucidates the pathobiological underpinnings of LTBI and holds promise for advancing diagnostic and prognostic applications [42]. An increasing number of studies have indicated that exosomes can transfer miRNAs to macrophages and regulate target genes. Variations in the exosomal miRNA content from plasma could serve as promising non-invasive biomarkers for the early identification of infectious states [43,44].

In conclusion, in this research, 12 differentially expressed miRNAs were identified in plasma-derived exosomes and confirmed via qRT-PCR using the same exosomes. Moreover, our results showed that plasma-derived exosomes promoted the intracellular survival of *Mtb* by transporting miR-7850-5p to directly inhibit SLC11A1 expression at the post-transcriptional level. Future studies should focus on identifying the underlying mechanisms of ferroptosis in the development of LTBI and TB.

## Figures and Tables

**Figure 1 microorganisms-12-01417-f001:**
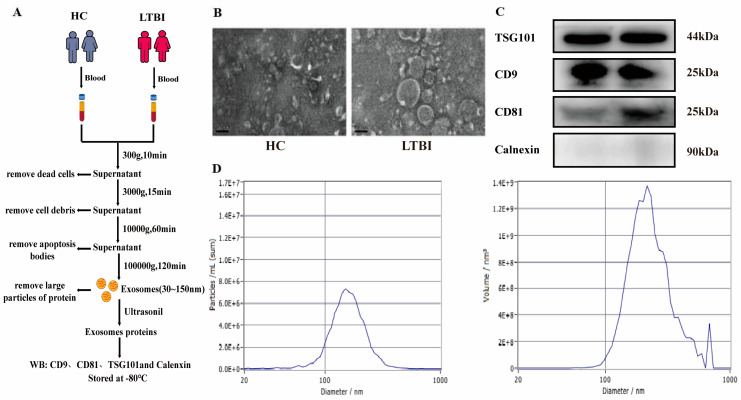
Characterization of the plasma-derived exosomes: (**A**) Schematic representation illustrating the process of isolating and performing a proteomic analysis on plasma-derived exosomes. (**B**) Transmission electron microscopy (TEM) analysis was performed to visualize the morphological structure of the isolated exosomes. (**C**) Western blotting analysis was carried out to assess the expression of exosomal markers, including TSG101, CD81, CD9, and calnexin, in the isolated exosomes. (**D**) NTA was conducted to determine the size distribution of the isolated exosomes.

**Figure 2 microorganisms-12-01417-f002:**
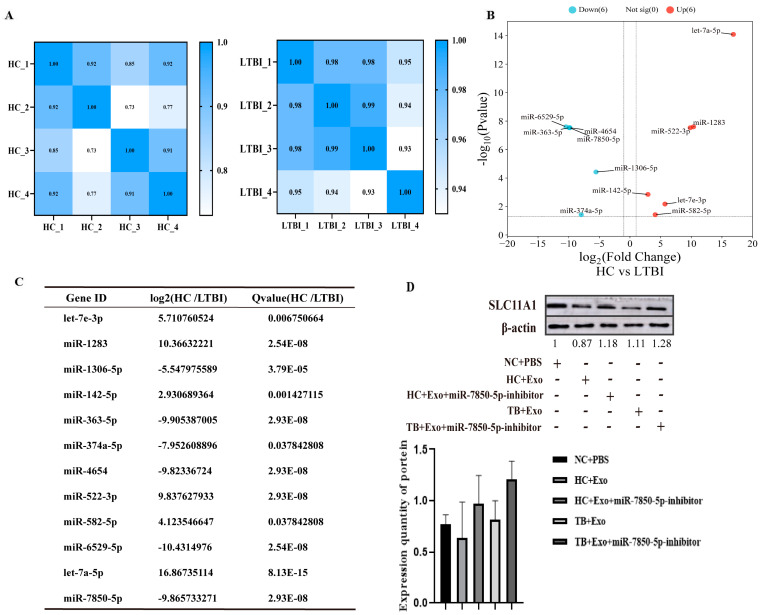
Exosomal DEmiRNAs identified by small RNA-Seq revealed 12 differentially expressed miRNAs in individuals with latent tuberculosis infection (LTBI) compared to healthy controls. (**A**) Assessment of correlation between biological replicates in the control group (n = 4) and plasma samples from individuals with latent tuberculosis infection (LTBI) (n = 4). (**B**) Visualization of differentially expressed exosomal DEmiRNAs, highlighting statistically significant upregulation (depicted by red points) and downregulation (depicted by blue points) in the control group (n = 3) compared to plasma samples from LTBI patients (n = 4) through a volcano plot. (**C**) The details of DEmiRNAs. (**D**) Exo-miR-7850-5p incubation in THP1 cells resulted in the downregulation of SLC11A1 expression, whereas miR-7850-5p inhibition led to an upregulation of SLC11A1 expression.

**Figure 3 microorganisms-12-01417-f003:**
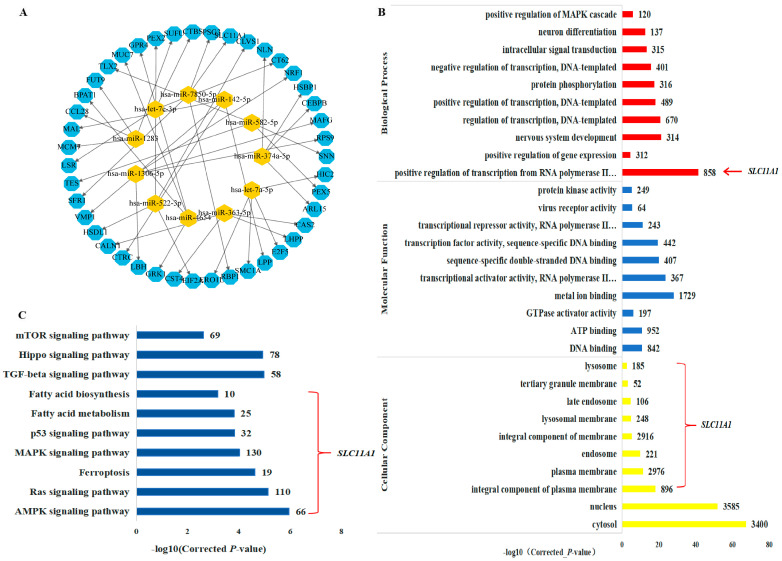
Functional analysis of exosomes differential miRNAs. (**A**) Network analysis of exosomal miRNAs with differential expression patterns and their corresponding target genes. (**B**) Gene Ontology enrichment visualization of exosomal miRNAs showing differential expression, encompassing biological processes, molecular functions, and cellular components. (**C**) Bubble chart depicting the KEGG pathway enrichment analysis of exosomal miRNAs exhibiting differential expression profiles.

**Figure 4 microorganisms-12-01417-f004:**
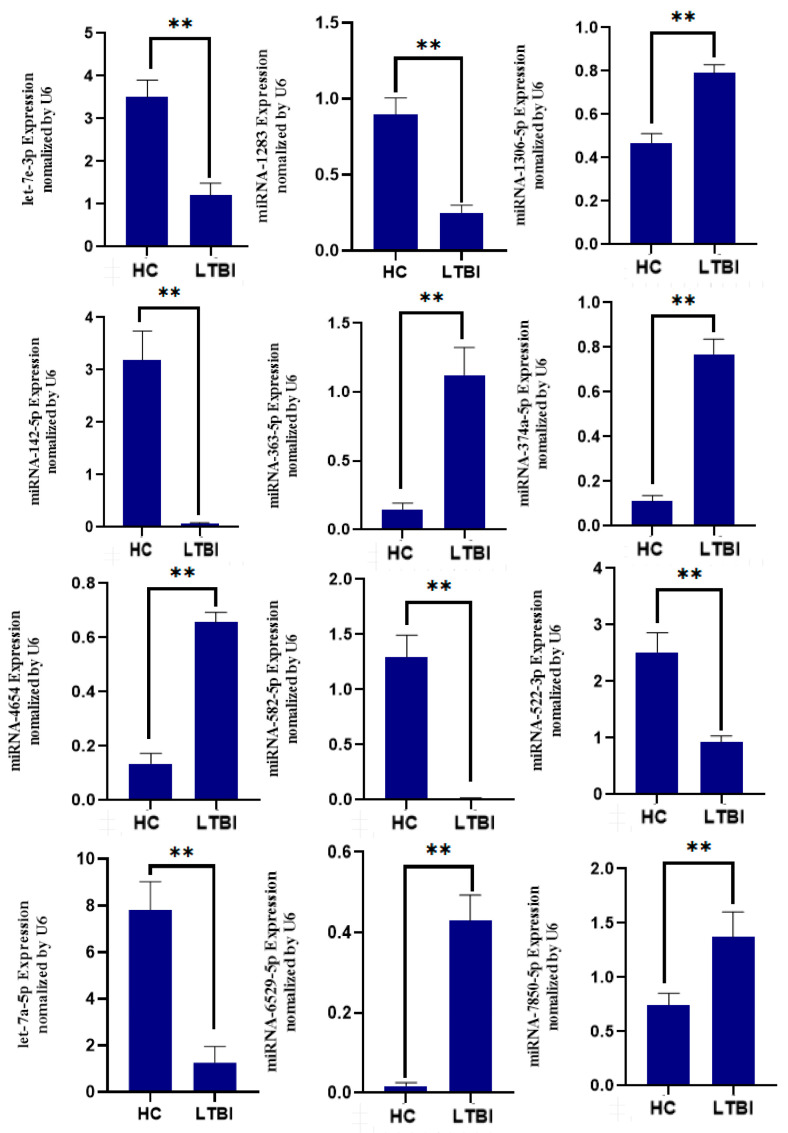
Validation of miRNA candidates from HC and LTBI exosomal samples. ** *p* < 0.01 for the expression of miRNAs in the LTBI compared to HC.

**Figure 5 microorganisms-12-01417-f005:**
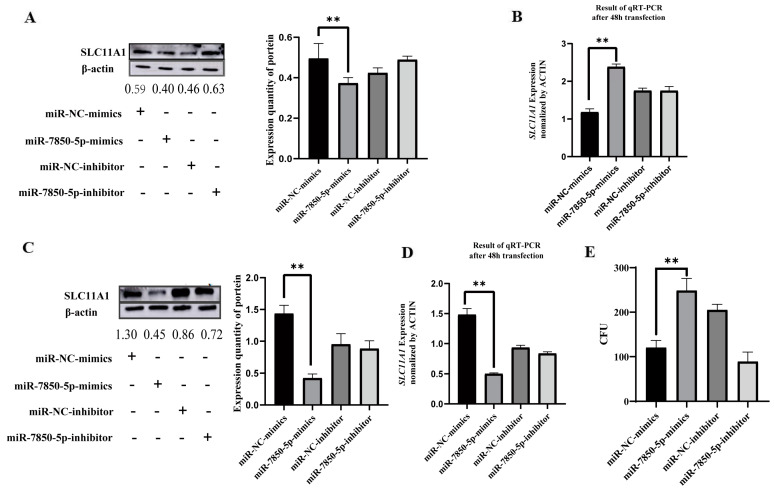
miR-7850-5p modulates the susceptibility to *Mycobacterium tuberculosis* by targeting SLC11A1. (**A**) Reduced SLC11A1 expression was observed in *Mtb*-uninfected THP1 cells upon transfection with miR-7850-5p mimics. (**B**) Transfection with miR-7850-5p mimics led to decreased mRNA levels of SLC11A1 in *Mtb*-uninfected THP1 cells. (**C**) Decreased SLC11A1 expression was observed in *Mtb*-infected THP1 cells following transfection with miR-7850-5p mimics. (**D**) Transfection with miR-7850-5p mimics resulted in the downregulation of SLC11A1 mRNA levels in *Mtb*-infected THP1 cells. (**E**) Intracellular survival of *Mtb* was assessed using the CFU assay. ** *p* < 0.01 for the expressions of SLC11A1 and miR-7850-5p in the group of miR-7850-5p mimics compared to miR-NC-mimics.

**Table 1 microorganisms-12-01417-t001:** Characteristics of the study population.

Characteristics	Total, n(%)	Healthy Controls, n(%)	LTBI Patients, n(%)	
**Age, mean ± SD, years** **Age Range**	HC: 34 LTBI: 16	32.00 ± 5.20 26–39	30.25 ± 8.33 24–48	
**Sex**				<0.01
Male	6(12%)	2(5.88%)	4(25%)	
Female	44(88%)	32(94.12%)	12(75%)	
**Smoking**				
Yes	5(10%)	1(2.94%)	4(25%)	
No	45(90%)	33(97.06%)	12(75%)	
**Drinking**				
Yes	10(20%)	8(23.53%)	2(12.5%)	
No	40(80%)	26(76.47%)	14(87.5%)	
**Education Level**				
Junior school or lower	0(0%)	0(0%)	0(0%)	
Senior high school and higher	50(100%)	34(100%)	16(100%)	
**Marital Status**				
Married	15(30%)	9(26.47%)	6(37.5%)	
Unmarried/divorced/widowed	35(70%)	25(73.53%)	10(62.5%)	

## Data Availability

The raw data supporting the conclusions of this article will be made available by the authors on request.

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
