# Peer review of "Exosomal Small RNA Sequencing Profiles in Plasma from Subjects with Latent Mycobacterium tuberculosis Infection"

_microorganisms, 2024, doi:10.3390/microorganisms12071417_

Round 1

Reviewer 1 Report

Comments and Suggestions for Authors

Please consider these revisions, a final revision for languaje, and properly cite and discuss the works of, Ly et al 2017 (10.3389/fmicb.2017.0105) and Lyu et al., 2019 (10.3389/fmicb.2019.01174)

Line 35. Use the same abbreviation for Mycobacterium tuberculosis (Mtb)

Lines 64 -71. The text specifies a total of 48 participants in the study, 32 LTBI (4 pools of 8) and 16 HC (4 pools of 4). However, the Table 1 shows the characteristics of a population of 50 participants (34 LTBI, 16 HC). Please solve this discrepancy in the total number of LTBI participants and the way the plasma pools were prepared with the 34 samples.

Lines 86-93. Please edit, verify and cite the methodology correctly. This paragraph is identical to that reported in www.ncbi.nlm.nih.gov | Small RNA Profiles of Serum Exosomes Derived From Individuals With Latent and Active Tuberculosis - PMC

https://www.ncbi.nlm.nih.gov/pmc/articles/PMC6546874/

Lines 94-98. The same, please edit, verify and cite the methodology correctly. This paragraph is identical to that reported in journal.frontiersin.org | Frontiers | RNA Profiling Analysis of the Serum Exosomes Derived from Patients with Active and Latent Mycobacterium tuberculosis Infection

http://journal.frontiersin.org/article/10.3389/fmicb.2017.01051/full

Line 103. Eliminate gel after (PAGE), the abbreviation is already explained

Lines 133-136. No quantities are specified for transfection with miR-7850-5p mimic or inhibitor ¿Are the same mentioned in section 2.8?

Line 137. Use transfected instead of infected.

Line 150. Eliminate of

Lines 152-161. No details are mentioned about in vitro infection with Mtb (strain, duration of infection, biosafety level used), nor details about the CFU assay, the reports are by mL? this is important for interpreting better the Figure 5E.

Figure 1D. Change remove apoptosis body for "remove apoptosis bodies".

Lines 203. About correlation coefficient, please correct to: among the four pools or samples if there are 16 individuals in group LTBI as explained in section 2.1.

Resolution of figure 2 is low, it needs to be improved.

Figure 2. Downregulated exosomal DEmiRNAs are shown in blue, and the figure caption indicates they are depicted by green points.

Table S2 appears titled as Profile of the bovine novel miRNAs. Please verify that it is correct.

Figure 3 also needs higher resolution.

Line 333. Please edit, all paper indicates that were 12 not 10, the differentially expressed miRNAs identified.

Comments on the Quality of English Language

A final revision for language is recommended as some errors are detected. Better if a certificate of edition exists

Author Response

Responses to Reviewer 1

Comments and Suggestions for Authors

Please consider these revisions, a final revision for languaje, and properly cite and discuss the works of, Ly et al 2017 (10.3389/fmicb.2017.0105) and Lyu et al., 2019 (10.3389/fmicb.2019.01174)

Lines 86-93. Please edit, verify and cite the methodology correctly. This paragraph is identical to that reported in www.ncbi.nlm.nih.gov | Small RNA Profiles of Serum Exosomes Derived From Individuals With Latent and Active Tuberculosis - PMC

https://www.ncbi.nlm.nih.gov/pmc/articles/PMC6546874/

Lines 94-98. The same, please edit, verify and cite the methodology correctly. This paragraph is identical to that reported in journal.frontiersin.org | Frontiers | RNA Profiling Analysis of the Serum Exosomes Derived from Patients with Active and Latent Mycobacterium tuberculosis Infection

http://journal.frontiersin.org/article/10.3389/fmicb.2017.01051/full

Response:Thank you very much for your comments. Our approach does align closely with the methods presented in these two articles. We have revised our method and cited these two articles as references. (please see Line 102-121)

Line 35. Use the same abbreviation for Mycobacterium tuberculosis (Mtb)

Response: Thanks so much for your suggestion. Done as requested (please see Line 44).

Line 103. Eliminate gel after (PAGE), the abbreviation is already explained

Response: Thanks so much for your suggestion. Done as request (please see Line 125).

Lines 133-136. No quantities are specified for transfection with miR-7850-5p mimic or inhibitor ¿Are the same mentioned in section 2.8?

Response: Thanks so much for your question. Indeed, the quantities required for transfection with the miR-7850-5p mimic or inhibitor are identical to those specified in section 2.8.

Line 137. Use transfected instead of infected.

Response: Done as request (Please see Line 162).

Line 150. Eliminate of

Response: Done as request (Please see Line 177).

Lines 152-161. No details are mentioned about in vitro infection with Mtb (strain, duration of infection, biosafety level used), nor details about the CFU assay, the reports are by mL? this is important for interpreting better the Figure 5E.

Response: Mtb H37Ra was provided by Prof. Qian Gao (Fudan University, Shanghai, China).

THP1 cells were infected with H37Ra at a multiplicity of infection (MOI) of 10:1 (bacteria to cell ratio) for 10 hours (P2+ biosafety level). Subsequently, the infected cells were washed three times with PBS in preparation for subsequent experiments. Transfected THP1 cells were infected with Mtb at an MOI of 10 for 10 hours and then lysed using 0.1% Triton-X 100 (Solarbio, China). Serial dilutions of 10-fold and 100-fold were prepared and utilized for quantitative culture. Subsequently, 60 μL of each dilution was plated on Middlebrook 7H10 agar plates (BD, USA) supplemented with 10% OADC enrichment (BD, USA). Following a 2-week incubation, Mtb colonies were observed, and colony-forming units (CFUs) were calculated (Please see Line 178-196).

Figure 1D. Change remove apoptosis body for "remove apoptosis bodies".

Response: Thanks so much for your suggestion. Done as requested (Please see Figure 1).

Lines 203. About correlation coefficient, please correct to: among the four pools or samples if there are 16 individuals in group LTBI as explained in section 2.1.

Response: Thanks so much for your suggestion. Done as requested (Please see Line 239-240).

Resolution of figure 2 is low, it needs to be improved.

Response: Thanks so much for your suggestion. When we insert images into the manuscript, the image pixels decrease. If allowed, we can upload the original high-resolution image as an attachment.

Figure 2. Downregulated exosomal DEmiRNAs are shown in blue, and the figure caption indicates they are depicted by green points.

Response: Thanks for reviewer's kind reminding. We have changed the figure caption (Please see Figure 2 caption).

Table S2 appears titled as Profile of the bovine novel miRNAs. Please verify that it is correct.

Response: Thanks for reviewer's kind reminding. We used a similar table for editing our results and forgot to modify the table header while editing. This was an error we made and have since rectified (Please see Table S2).

Figure 3 also needs higher resolution.

Response: Thanks so much for your suggestion. When we insert images into the manuscript, the image pixels decrease. If allowed, we can upload the original high-resolution image as an attachment.

Line 333. Please edit, all paper indicates that were 12 not 10, the differentially expressed miRNAs identified.

Response: Thanks for your kind reminding. We have checked the errors of the manuscript and revised them. After confirmation, we identified a total of 12 differential expressions of miRNA through plasma exosome small RNA sequencing.

Comments on the Quality of English Language

A final revision for language is recommended as some errors are detected. Better if a certificate of edition exists

Response: Thanks for reviewer's kind reminding. We have perused the content and modified the wording several rounds with language corrections using the English polishing service recommended by MDPI. 

Reviewer 2 Report

Comments and Suggestions for Authors

Review Cui et al, “Exosomal Small RNA Sequencing Profiles in Plasma from 2

Subjects with Latent Mycobacterium Tuberculosis Infection”

This study compares the miRNA contained in exosomes in healthy controls and people latently infected with M. tuberculosis Mtb.  The idea of looking at exosomes as reflecting the miRNAs as reflecting something about immune status is interesting.  The authors propose that their results support show that latently TB infected (LTBI) individuals have 12 differentially expressed miRNAs compared to healthy controls, and then focus on miR-7850-5p.  further, they propose that higher miR 7850 – 5p inhibits expression of SLC11A and this leads to increased survival of Mtb in macrophages.  Overall, the results are not convincing and the speculation on the roles of the miRNAs is not supported by any data, except increased survival of Mtb in macrophages.  However, people with LTBI control their infection, apparently not allowing significant growth of the MTB, so the conclusion is puzzling.

Abstract:

Line 9 Current estimates are 23% of global population is latently infected.

Line 35. Is “MT” Mtb?

Lines 37 – 41 (and the abst) talk about the need to diagnose LTBI, but this is not mentioned in the results.

Lin 56 Rather than “selective sorting”, the results suggest that the Exo package reflects what is more highly present in the cell.

Line 78.  What were the “ comparable biomarkers”?

Line 93 negative staining

Lines 135 – 6.  What were the miR-7850-5p mimics and inhibitors?

Line 158 PMA is not identified.

Line 174-5. The sample is not adequate to conclude anything about an association of gender and latent TB.

Line 182 TEM is only defined in the figure legend.

Lines 185 –markers CD9, CD63, and TSG101, but figure 1B shows CD9, CD81 and TSG101, NO CD63.

Figure 1D. What is “Ultrasonil”? What is PTS digestion?

Line 202 What is Q20? What is CPM?

Line 204 concordance of only 3 out of 4 pools is not very convincing.

Figure 2D From the Western blots, the intensity of the SLC11A1 band in the NC-PBS lane appears much stronger than either TB-Exo or TB exo miR-7850 5p inhibitor lanes, but this is the opposite of what is shown in the graph. The relative intensity of the b-actin bands do not appear to explain this. The Std Dev. Bars overlap considerably between HC-Exo and HC Exo miR inhibitor and NC-PBS, and there are no statistics given to show a significant difference.  How many times was this done to create the Std. Dev bars?

2D?  Exo were incubated with the cells, but the Exo’s contained several RNAs, so it is not clear that miR-7850-5p is causing the difference, which is not, anyway, convincing.  The level of SLC11A1 in the TB-Exo is the same as NC-PBS, without evidence of reduction.

2C – How many assays does this represent?  There are no statistics to evaluate the significance of the differences.

Line 248 – 259.and Figure 4. How many samples were tested to evaluate the DEmiRNAs and obtain the significance, and what statistic was used? All the “functions” or pathways are unproven, so comments on their role is speculation.

Figure 5:  How many times was this performed? On all the pools?  What statistic was used to calculate significance?

5A, The graph does not show increased protein with the inhibitors compared to the NC-mimics

5B  The SLC11A qRT-PCR is greater with the mimics, but the legend says  SLC11A expression is decreased with the transfected mimics.

5C & 5D The levels of SLC11A protein and RNA are higher in the NC-mimic control than with the inhibitor, and the miR 7850 -5p inhibitor is the same as the control NC-inhibitor.

5E Why should the miR-NC inhibitor provoke higher intracellular survival than the NC mimic? 

These figures show unconvincing results that contradict the proposal that miR-7850 mimics reduce SLC11A expression and thereby increase Mtb intracellular survival.

Discussion The roles of the DEmiRNAs derived from bioinformatics are speculation and have no proof.  It is not clear why LTBI individuals would have miRNA’s that promote the survival of Mtb in macrophages. 

Overall, analysis of miRNAs in exosomes is interesting, but the results are far from convincing and the significance is mere speculation.   A more complete study is needed, and results must be consistent.  Speculation on the role of any DEmiRNAs require some proof before they can have any meaning.

Table S2 – Why are they BOVINE novel miRNAs

Comments on the Quality of English Language

The language has a few errors, but overall is not bad.  However, explanations could be more developed, as the style is minimal in some places.

Author Response

Responses to Reviewer 2

Comments and Suggestions for Authors

Review Cui et al, “Exosomal Small RNA Sequencing Profiles in Plasma from 2 Subjects with Latent Mycobacterium Tuberculosis Infection”

This study compares the miRNA contained in exosomes in healthy controls and people latently infected with M. tuberculosis Mtb.  The idea of looking at exosomes as reflecting the miRNAs as reflecting something about immune status is interesting.  The authors propose that their results support show that latently TB infected (LTBI) individuals have 12 differentially expressed miRNAs compared to healthy controls, and then focus on miR-7850-5p.  further, they propose that higher miR 7850 – 5p inhibits expression of SLC11A and this leads to increased survival of Mtb in macrophages.  Overall, the results are not convincing and the speculation on the roles of the miRNAs is not supported by any data, except increased survival of Mtb in macrophages.  However, people with LTBI control their infection, apparently not allowing significant growth of the MTB, so the conclusion is puzzling.

Abstract:

Line 9 Current estimates are 23% of global population is latently infected.

Response: Thanks so much for your suggestion. Done as requested. (Please see Line 14)

Line 35. Is “MT” Mtb?

Response: Thanks so much for your suggestion. Done as requested. (Please see Line 44)

Lines 37 – 41 (and the abst) talk about the need to diagnose LTBI, but this is not mentioned in the results.

Response: Thanks so much for your comments. The present study aims to conduct cell-level validation of the screened potential miRNAs and their target genes, providing a basis for the diagnosis of latent infection. Further in-depth research is still required for the treatment of latent infection, and the current results of this study cannot draw any conclusions regarding the treatment of latent infection.

Lin 56 Rather than “selective sorting”, the results suggest that the Exo package reflects what is more highly present in the cell.

Response: Thanks so much for your comments. Your suggestions are extremely important to us. We have carefully considered them and made the changes here in accordance with your recommendations (Please see Line 73).

Line 93 negative staining

Response: Thanks so much for your suggestion. Done as requested (Please see Line 114).

Lines 135 – 6.  What were the miR-7850-5p mimics and inhibitors?

Response: Thanks so much for your suggestions. miRNA mimics are double-stranded small RNA molecules designed based on the mature miRNA sequence, used to mimic the endogenous mature miRNA sequence. miRNA inhibitor is a tool used to suppress the function of endogenous miRNA.

The functional of miRNA mimics is as follows:

miRNA mimics can be introduced into cells expressing the corresponding miRNA, mimicking the function of microRNAs, or can be used in conjunction with a dual-luciferase reporter system containing miRNA binding sites to verify the regulatory relationship between miRNAs and their target genes.

The functional mechanism of miRNA inhibitors is as follows:

  1. Effectively inhibit the function of endogenous mature miRNA.
  2. Design 21-23 nt 2'-O-methyl modified RNA oligonucleotides.
  3. Low concentrations can efficiently and persistently inhibit miRNA activity.

Line 158 PMA is not identified.

Response: Thanks so much for your comments. Done as requested (Please see Line 158).

Line 174-5. The sample is not adequate to conclude anything about an association of gender and latent TB.

Response: Thanks so much for your comments. According to the references, it is currently not possible to determine a significant association between gender and the incidence rate of latent tuberculosis. In our current study, we compared and analyzed the demographic characteristics of 34 HC and 16 LTBI patients, and the results showed that the gender difference was statistically significant (p<0.01). However, due to our limited sample size, we cannot draw significant conclusions on the gender-related incidence of tuberculosis, and further sample expansion is needed for research.

Line 182 TEM is only defined in the figure legend.

Response: Thanks so much for your kind reminding. TEM were mentioned in Materials and Methods (Please see Line 106-115).

Lines 185 –markers CD9, CD63, and TSG101, but figure 1B shows CD9, CD81 and TSG101, NO CD63.

Response: Thanks so much for your suggestion. This is our mistake, we did CD81, but it was mistakenly written as CD63. This has been changed (Please see the Revised manuscript).

Figure 1D. What is “Ultrasonil”? What is PTS digestion?

Response: Thanks for your questions. Both Ultrasonil and PTS digestion are preparatory works for extracting exosomal proteins. Perform ultrasonic digestion on our separated exosomes to extract proteins, preparing for subsequent detection of exosomal protein markers. We made slight adjustments to Figure 1A, please refer to Figure 1A.

Line 202 What is Q20? What is CPM?

Response: Thanks so much for your comments.

Q20 is the proportion of bases with a phred base quality score ≥20; i.e., the proportion of read bases with error rates<1%.

The expression of miRNA expression was measured as counts per million (CPM) using the following formula: normalized expression = mapped read count/total reads×1000000.

References:

Zhou, L., Chen, J., Li, Z., Li, X., Hu, X., Huang, Y., et al. (2010). Integrated profiling of microRNAs and mRNAs: microRNAs located on Xq27.3 associate with clear cell renal cell carcinoma. PLoS One 5, e15224.

Anders, S., and Huber, W. (2010). Differential expression analysis for sequence count data. Genome Biol 11, R106.

Line 204 concordance of only 3 out of 4 pools is not very convincing.

Response: Thanks so much for your questions. The correlation coefficients between HC2 vs HC3, and HC2 vs HC4, are 0.73 and 0.77, respectively. While these correlations are not particularly low, given that we have a total of four replicates, to ensure the credibility of the experimental results, we have excluded this replicate.

Figure 2D From the Western blots, the intensity of the SLC11A1 band in the NC-PBS lane appears much stronger than either TB-Exo or TB exo miR-7850 5p inhibitor lanes, but this is the opposite of what is shown in the graph. The relative intensity of the b-actin bands do not appear to explain this. The Std Dev. Bars overlap considerably between HC-Exo and HC Exo miR inhibitor and NC-PBS, and there are no statistics given to show a significant difference.  How many times was this done to create the Std. Dev bars?

2D?  Exo were incubated with the cells, but the Exo’s contained several RNAs, so it is not clear that miR-7850-5p is causing the difference, which is not, anyway, convincing.  The level of SLC11A1 in the TB-Exo is the same as NC-PBS, without evidence of reduction.

2C – How many assays does this represent?  There are no statistics to evaluate the significance of the differences.

Response: Thanks so much for your comments. In this study, all experiments were conducted at least three times. Transfection of miR-7850-5p inhibitors suppressed the endogenous expression of miR-7850-5p in cells, leading to an increase in SLC11A1 protein expression. Compared to the NC group, the results showed that the miRNA-7850-5p in the exosomes may be released into the recipient cells and regulate the expression of SLC11A1.

Line 248 – 259.and Figure 4. How many samples were tested to evaluate the DEmiRNAs and obtain the significance, and what statistic was used? All the “functions” or pathways are unproven, so comments on their role is speculation.

Response: Thanks so much for your kind reminding. In this study, we collected plasma samples from 34 healthy individuals and 16 latently infected individuals. For pooling, 8 plasma samples from healthy individuals were pooled together, resulting in a total of 32 pooled plasma samples, while 4 plasma samples from latently infected individuals were pooled together, resulting in a total of 16 pooled plasma samples. Extracellular vesicles were then isolated from the pooled plasma samples using differential centrifugation, and total RNA was extracted for sequencing. The remaining samples from the sequencing process were subsequently used for RT-qPCR validation. Through pathway analysis, we have screened miR-7850-5p and its target gene SLC11A1, and have verified them at the cellular level. We have not conducted research on all pathways, and we will further screen and verify them in the future.

Figure 5:  How many times was this performed? On all the pools?  What statistic was used to calculate significance?

5A, The graph does not show increased protein with the inhibitors compared to the NC-mimics

5B The SLC11A qRT-PCR is greater with the mimics, but the legend says SLC11A expression is decreased with the transfected mimics.

5C & 5D The levels of SLC11A protein and RNA are higher in the NC-mimic control than with the inhibitor, and the miR 7850 -5p inhibitor is the same as the control NC-inhibitor.

5E Why should the miR-NC inhibitor provoke higher intracellular survival than the NC mimic?

These figures show unconvincing results that contradict the proposal that miR-7850 mimics reduce SLC11A expression and thereby increase Mtb intracellular survival.

Response: Thanks so much for your questions. Figure 5 verifies the regulatory relationship between miR-7850-5p and SLC11A1 in the THP1 cell line, as well as the regulatory relationship between miR-7850-5p and SLC11A1 after simulating Mtb infection in macrophages. Each experiment was repeated at least three times. The results of the miR-7850-5p mimic were compared with the mimic NC, and the results of the miR-7850-5p inhibitor were compared with the inhibitor NC. Since miRNAs are post-transcriptional regulators, they mainly affect the protein level of target genes, and the changes in mRNA expression are not very obvious.

Discussion The roles of the DEmiRNAs derived from bioinformatics are speculation and have no proof.  It is not clear why LTBI individuals would have miRNA’s that promote the survival of Mtb in macrophages.

Overall, analysis of miRNAs in exosomes is interesting, but the results are far from convincing and the significance is mere speculation.   A more complete study is needed, and results must be consistent.  Speculation on the role of any DEmiRNAs require some proof before they can have any meaning.

Table S2 – Why are they BOVINE novel miRNAs

Response: Thanks for your kind reminding. We have modified it (Please see Table S2).

Comments on the Quality of English Language

The language has a few errors, but overall is not bad.  However, explanations could be more developed, as the style is minimal in some places.

Response: Thanks for reviewer's kind reminding. We have perused the content and modified the wording several rounds with language corrections using the English polishing service recommended by MDPI.

Reviewer 3 Report

Comments and Suggestions for Authors

1. What was the reason for mixing plasma samples from different donors since this may not provide specific information on the exosomes in each individual donor belonging to the two groups?

2. Methods section needs complete reorganization to achieve a logical flow.

Consider putting all THP-1 studies together to make it more cohesive.

3 It is unclear how the authors determine higher incidence of latent TB among genders

4. Discussion section should be elaborated to include details on the background information and literature reports in support of the study findings. More in-depth specifics should be included about how exosomes alter metabolic and physiological functions.

5. The manuscript should be edited to improve both the language and quality of the scientific writing.

Comments on the Quality of English Language

Manuscript should be edited to enhance the overall clarity.

Author Response

Responses to Reviewer 3

Comments and Suggestions for Authors

  1. What was the reason for mixing plasma samples from different donors since this may not provide specific information on the exosomes in each individual donor belonging to the two groups?

Response: Thanks so much for your comments. The purpose of our mixed pool sequencing is to increase the sample size and ensure the accuracy of the sequencing results. After reviewing a large number of references, we found that many articles use this method for sequencing. Several related references were listed as follows. A total of 90 serum samples were collected in this paper The serum samples were grouped according to the clinical cohort as ATB, LTBI, and healthy control (HC). Serum was obtained from each participant and then pooled based on the group (pooled n D 15 for HC, LTBI, and ATB, respectively; 1 mL of serum in each pooled sample) (Lv L et al. RNA Profiling Analysis of the Serum Exosomes Derived from Patients with Active and Latent Mycobacterium tuberculosis Infection. Front Microbiol.).

In another paper, Zhang Min et al. collected a total of blood samples from 55 ATB patients and 45 healthy people, and extracted exosomes after mixing all serum of TB and HC group (Zhang M et al. Proteomics Analysis of Exosomes From Patients With Active Tuberculosis Reveals Infection Profiles and Potential Biomarkers. Front Microbiol.).

We would like to hear more instructive comments.

References:

Lv L, Li C, Zhang X, Ding N, Cao T, Jia X, Wang J, Pan L, Jia H, Li Z, Zhang J, Chen F, Zhang Z. RNA Profiling Analysis of the Serum Exosomes Derived from Patients with Active and Latent Mycobacterium tuberculosis Infection. Front Microbiol. 2017 Jun 12; 8:1051.

Zhang M, Xie Y, Li S, Ye X, Jiang Y, Tang L, Wang J. Proteomics Analysis of Exosomes From Patients With Active Tuberculosis Reveals Infection Profiles and Potential Biomarkers. Front Microbiol. 2022 Jan 6;12: 800807.

Han J, Cui X, Yuan T, Yang Z, Liu Y, Ren Y, Wu C, Bian Y. Plasma-derived exosomal let-7c-5p, miR-335-3p, and miR-652-3p as potential diagnostic biomarkers for stable coronary artery disease. Front Physiol. 2023 May 9;14: 1161612.

  1. Methods section needs complete reorganization to achieve a logical flow.

Consider putting all THP-1 studies together to make it more cohesive.

Response: Thanks so much for your suggestion. Your suggestions are very important to us, and we have also thought about how to arrange the order of the article. Our principle of arrangement is as follows, please refer to: In the part of materials and methods, sample preparation was firstly followed by exosome isolation and RNA extraction from the sample, and then small RNA was prepared for RNA sequencing, and the sequencing results were analyzed, and then miRNA expression was compared by RT-qRCR, and cell experiments were conducted on THP1.

On THP1 cells, we first incubated the extracted exosomes with THP1 cells. Western blot analysis demonstrated that the expression of SLC11A1 was inhibited upon incubation with exo-miR-7850-5p in THP1 cells, whereas inhibition of miR-7850-5p promoted the expression of SLC11A1. On this basis, the effect of miRNA-7850-5p on the expression of SLC11A1 was verified in the Mtb-uninfected and Mtb-infected THP1 cells.

3 It is unclear how the authors determine higher incidence of latent TB among genders                

Response: Thanks so much for your comments. According to the references, it is currently not possible to determine a significant association between gender and the incidence rate of latent tuberculosis. In our current study, we compared and analyzed the demographic characteristics of 34 HC and 16 LTBI patients, and the results showed that the gender difference was statistically significant (p<0.01). However, due to our limited sample size, we cannot draw significant conclusions on the gender-related incidence of tuberculosis, and further sample expansion is needed for research.

  1. Discussion section should be elaborated to include details on the background information and literature reports in support of the study findings. More in-depth specifics should be included about how exosomes alter metabolic and physiological functions.

Response: Thanks so much for your suggestions. The specific details of how extracellular vesicles exert their physiological functions and relevant references have been added in the discussion section (Please see Line 328-338).

  1. The manuscript should be edited to improve both the language and quality of the scientific writing.

Response: Thanks for reviewer's kind reminding. We have perused the content and modified the wording several rounds with language corrections using the English polishing service recommended by MDPI.

Reviewer 4 Report

Comments and Suggestions for Authors

Introduction:
The number of LTBI (latent tuberculosis infection) cases worldwide, including the statement that approximately 10% of these cases can become active, must be provided. It's crucial to outline the associated risk factor

Additionally, it's essential to specify the name of the Dx on line 38.

highlighting that strategies employed are primarily focused on vulnerable populations is important.

To enhance understanding, provide a more detailed description of exosomes and their significance, including the markers CD9, CD63, and TSG101

Materials and Methods:

Is this protocol approved by the bioethics committee? Mentioning the name of the registry, along with describing the collection location and the parameters of the selected LTBI, is crucial.

Regarding line 158, it may be more advantageous to specify the cell counts differentiated by assay, whether they were stimulated in a well, in a plate, or through another method.

On line 160, specify which bacteria were added.

Results:
Figure 1 now appropriately includes the size of WB (Western Blot) markers, incorporating MPM (membrane protein marker), and Figure D has been moved to the beginning of the figure.

For Figure 2, ensure to describe the labels, particularly for (D).

Regarding line 234, further elucidate on the inhibition of miR-7850-5p. Provide details on the mechanisms or effects associated with inhibiting miR-7850-5p to enhance clarity.

Author Response

Responses to Reviewer 4

Comments and Suggestions for Authors

Introduction:

The number of LTBI (latent tuberculosis infection) cases worldwide, including the statement that approximately 10% of these cases can become active, must be provided. It's crucial to outline the associated risk factor

Additionally, it's essential to specify the name of the Dx on line 38.

highlighting that strategies employed are primarily focused on vulnerable populations is important.

To enhance understanding, provide a more detailed description of exosomes and their significance, including the markers CD9, CD63, and TSG101

Response: Thanks so much for your comments. The specific details of how extracellular vesicles exert their physiological functions and relevant references have been added in the introduction and discussion section (Please see Line 55-61 and 322-338).

Materials and Methods:

Is this protocol approved by the bioethics committee? Mentioning the name of the registry, along with describing the collection location and the parameters of the selected LTBI, is crucial.

Response: Thanks so much for your questions. Please see the part of Ethics Statement (Line 394-397).

Regarding line 158, it may be more advantageous to specify the cell counts differentiated by assay, whether they were stimulated in a well, in a plate, or through another method.

On line 160, specify which bacteria were added.

Response: Thanks so much for your comments. We have re-edited this part (Please see Line 178-196).

Results:

Figure 1 now appropriately includes the size of WB (Western Blot) markers, incorporating MPM (membrane protein marker), and Figure D has been moved to the beginning of the figure.

Response:Thanks so much for your suggestion. Done as rerquested (Please see Figure 1).

For Figure 2, ensure to describe the labels, particularly for (D).

Response: Thanks for reviewer's kind reminding. Downregulated exosomal DEmiRNAs are shown in blue, and we have changed the figure caption (Please see Figure 2 caption).

Regarding line 234, further elucidate on the inhibition of miR-7850-5p. Provide details on the mechanisms or effects associated with inhibiting miR-7850-5p to enhance clarity.

Response: Thanks so much for your suggestions.

The functional mechanism of miRNA inhibitors is as follows:

  1. Effectively inhibit the function of endogenous mature miRNA.
  2. Design 21-23 nt 2'-O-methyl modified RNA oligonucleotides.
  3. Low concentrations can efficiently and persistently inhibit miRNA activity.

Round 2

Reviewer 2 Report

Comments and Suggestions for Authors

Review of revised version Cui et al, “Exosomal Small RNA Sequencing Profiles in Plasma from Subjects with Latent Mycobacterium Tuberculosis Infection”

While some of the minor problems and the English were corrected, there remain serious problems.

There are no line numbers for reference in the revised versions

Abst. “is a predominant fatal infectious disease globally” TB is not predominantly a fatal disease.  A small percentage of TB patients succumb to their illness.

In the introduction:

The two sentence beginning with “Exosomes contain…” Seem too much detail without explanation.

“Notably, exosomal micronRNAs have emerged as pivotal regulators…” A reference is needed to support this statement, as it is not supported by the references cited after the next sentence.

There is still no definition of “comparable biomarkers”

The composition of neither the miRNAs nor the inhibitors are detailed.

The statistical tests used are not stated.

The relative intensity of the bands in the western blots in Figures 2D and  5 A & C  do not appear to be reflected in the graphs.  This undermines the basic conclusions of the study. 

Why was SLC11A1 examined? This does not appear to be explained.

Q20 and CPM are not defined in the text.

It is still not clear why LTBI infected patients should have protein expression that would promote Mtb growth in macrophages, when these patients are apparently controlling the infection and MTb does not appear to be replicating.

The discussion is too long and rambling.

Comments on the Quality of English Language

The English is better.

Author Response

  1. Abst. “is a predominant fatal infectious disease globally” TB is not predominantly a fatal disease. A small percentage of TB patients succumb to their illness.

Response: Thanks so much for your insightful comments. Your advice is very professional and has greatly contributed to the improvement of our article. After consulting relevant professional literature, such as WHO's Global Tuberculosis Report, we have found that this sentence does have some issues. We have rephrased it accordingly. (Please see Line 12)

In the introduction:

  1. The two sentence beginning with “Exosomes contain…” Seem too much detail without explanation.

Response:Thank you for your insightful comments and suggestions. To facilitate the readers' understanding of exosomes, Reviewer 4 suggested that we introduce these biomarkers of exosomes. We thought this suggestion was very good, so we added these few sentences of introduction in this section.

  1. “Notably, exosomal microRNAs have emerged as pivotal regulators…” A reference is needed to support this statement, as it is not supported by the references cited after the next sentence.

Response: Thanks so much for your kind reminding. Done as suggested (Please see Line 60 and Ref. 9).

  1. There is still no definition of “comparable biomarkers”

Response:Thank you for your insightful comments and suggestions. LTBI is a condition where a person is infected with Mycobacterium tuberculosis but does not currently have ATB disease. Diagnosis of LTBI is typically made through positive results on tests like the tuberculin skin test (TST) or interferon-gamma release assay (IGRA). In individuals with LTBI, they may not show any symptoms and their immune system typically keeps the infection in check. Biomarkers for LTBI are often compared to those of healthy controls to differentiate between active TB and latent infection. In cases where the biomarkers of individuals with LTBI are comparable to those of healthy controls, it might indicate that the infection is not active and the immune system is effectively controlling the bacteria. I apologize for any confusion caused and have removed the incorrect statement.

  1. The composition of neither the miRNAs nor the inhibitors are detailed.

Response: Thanks so much for your kind reminding. Done as suggested (Please see Line 195-198).

  1. The statistical tests used are not stated.

Response: Thanks so much for your kind reminding. Done as suggested (Please see Line 201-207).

  1. The relative intensity of the bands in the western blots in Figures 2D and 5 A & C do not appear to be reflected in the graphs. This undermines the basic conclusions of the study.

Response: Thanks so much for your kind remind. In this current study, we repeated each group of western blotting at least three times, and the bar graph represents the statistics of the three repetitions. The suggestions you provided to us were very good, and we have annotated the expression levels of the proteins. (Please see Fig. 2D and 5A & 5C)

  1. Why was SLC11A1 examined? This does not appear to be explained.

Response: Thanks so much for your comments. Previous genome-wide association studies suggested that single-nucleotide polymorphisms (SNPs) in ASAP1/SLC11A1/SP110/TLR2/TLR4 could indicate the susceptibility to TB in some populations. In our previous study, we studied the association of SNPs in the immunity-related genes, i.e., ASAP1, SLC11A1, TLR2, TLR4 and SP110 genes with the susceptibility to TB in a Mongolian population in China. The results suggested that SLC11A1 showed potential association with an increased risk of TB and larger scale may help for further validation. (Han J, Cui XG, et al. Association of polymorphisms of innate immunity-related genes and tuberculosis susceptibility in Mongolian population. Hum Immunol. 2021;82(4):232-239.) In fact, we not only investigated the regulatory mechanism of miR-7850-5p and SLC11A1, but also conducted experiments on other potential miRNAs (miR-302a, miR-302e, and miR-372) of SLC11A1, unfortunately, the results were not very satisfactory. Therefore, the study may provide contribute to the identification of potential molecular targets for the detection and diagnosis of latent tuberculosis.

  1. Q20 and CPM are not defined in the text.

Response: Thanks so much for your kind reminding. Done as suggested (Please see Line 242-245).

  1. It is still not clear why LTBI infected patients should have protein expression that would promote Mtb growth in macrophages, when these patients are apparently controlling the infection and Mtb does not appear to be replicating. The discussion is too long and rambling.

Response: Thanks so much for your insightful comments. We fully agree with your comments. After reviewing the literature, it is indeed not clear why LTBI infected individuals have increased expression of proteins that promote the growth of Mtb in macrophages. In our current study, we found that miR-7850-5p regulates the expression of SLC11A1, and may inhibit the burden of Mtb by regulating the expression of SLC11A1. These will be the focus of our subsequent research. Thank you very much for your comments and suggestions. We hope that through your suggestions, we can achieve good research results. Moreover, we have perused the content and modified the wording several rounds with language corrections using the English polishing service recommended by MDPI and also checked by a colleague fluent in English writing.

Reviewer 3 Report

Comments and Suggestions for Authors

The authors have incorporated all the recommendations of the reviewers in the revision.

Comments on the Quality of English Language

Moderate editing of english language is required.

Author Response

Response:

Thanks for reviewer's kind reminding. We have perused the content and modified the wording several rounds with language corrections using the English polishing service recommended by MDPI.